# Polymeric Biocomposite Based on Thermoplastic Polyurethane (TPU) and Protein and Elastomeric Waste Mixture

**DOI:** 10.3390/ma16155279

**Published:** 2023-07-27

**Authors:** Mihaela Nituica, Ovidiu Oprea, Maria Daniela Stelescu, Maria Sonmez, Mihai Georgescu, Laurentia Alexandrescu, Ludmila Motelica

**Affiliations:** 1National Research and Development Institute for Textile and Leather, Division Leather and Footwear Institute, 93 Ion Minulescu Street, 031215 Bucharest, Romania; mihaelavilsan@yahoo.com (M.N.); dmstelescu@yahoo.com (M.D.S.); ficaimaria@yahoo.com (M.S.); mihai.georgescu@icpi.ro (M.G.); laurentia.alexandrescu@icpi.ro (L.A.); 2Faculty of Chemical Engineering and Biotechnologies, University Politehnica of Bucharest, 1-7 Gheorghe Polizu Streert, 011061 Bucharest, Romania; motelica_ludmila@yahoo.com; 3Academy of Romanian Scientists, Ilfov Street 3, 050044 Bucharest, Romania

**Keywords:** TPU, polymeric biocomposite, protein and elastomeric waste, mechanical properties, compatibilized

## Abstract

Polymeric biocomposites based on TPU/recycled TPUW/mixed leather and SBR rubber waste unmodified/modified with polydimethylsiloxane/PE-g-MA in different percentages were made via the mixing technique on a Plasti-Corder Brabender mixer with an internal capacity of 350 cm^3^. The waste, which came from the shoe industry, was cryogenically ground with the help of a cryogenic cyclone mill at micrometric sizes and different speeds. For the tests, standard plates of 150 × 150 × 2 mm were obtained in a laboratory-scale hydraulic press via the method of compression between its plates, with well-established parameters. The biocomposites were tested physico-mechanically and rheologically (MFI) according to the standards in force on polymer-specific equipment, also via FT-IR spectroscopy and microscopy, as well as via differential scanning calorimetry—DSC. Following the tests carried out, according to the standard for use in the footwear industry, at least two samples present optimal values (of interest) suitable for use in the footwear industry by injection or pressing in forming moulds.

## 1. Introduction

Recycling and reusing waste (technological and post-consumer waste), both at the European level and nationally, is deficient [1]. At the same time, due to non-biodegradability, but also to the growing consumption, the elimination of polymeric waste of any kind creates serious economic and environmental concerns, and waste management is thus becoming a major problem [1,2]. Taking into account the degree of awareness of what the environment means in today’s society, the most viable option remains recycling [3,4]. Thus, a series of laws and regulations were issued that take into account the management of waste from any field [5,6]. By applying these laws and regulations (regarding the reuse and recycling of waste, reducing of pollution), using modern and efficient technologies, and transforming waste into new products with added value, we can contribute to the protection of the environment, to the protection of human health and, last but not least, to the increased turnover of economic agents, ultimately leading to a circular and sustainable economy [7,8,9]. Also, by applying the principle of the 4Rs—reduction, repair, reuse and recycling of waste—we can have a healthy natural environment by eliminating waste from both the processing and post-consumer stage in the fields of textiles, leather, footwear, etc., as well as by reducing the carbon footprint, through which a large amount of CO_2_ can be eliminated, if the organic matter obtained from the waste is used for the production of polymeric biocomposites [1,7,8,9,10].

As conventional polymer materials are non-biodegradable, in order to obtain biocomposites, the new trends are to use natural materials, such as protein fibres, cellulose, etc., vulcanized elastomers and others as reinforcement materials [11,12]. Natural fibres can successfully replace reinforcing agents, generally inorganic reinforcing agents, such as carbon black, silicon, etc. [12]. Besides having the advantage of being low cost, non-toxic (both for the environment and for the human factor), easy to procure and biodegradable, they also improve the physical–mechanical properties of the composites (hardness, tensile strength, etc.) [12], but they also exhibit a non-abrasive behaviour on processing equipment [11,12,13,14,15,16,17].

The potential for recycling and reuse of technological or post-consumer waste, such as elastomeric and protein waste, is poorly exploited [18,19]. Protein waste is easy to recycle, does not maintain combustion, has self-extinguishing capacity and has a hygroscopicity of up to 37%, while elastomeric waste improves the hardness, elasticity, but also the resistance, and last but not least, by recycling and reusing them, the impact on the environment is greatly diminished [15,17,18,19,20,21,22].

Thermoplastic polyurethane (TPU) is used in the footwear industry: soles, heels, inserts for leather shoes, etc. It is resistant to abrasion, at low temperatures, resistant to aggressive working environments, has adhesion to the surface and is slip resistant, and returns to its shape when it is deformed, and the working temperature is relatively low—80 °C [23,24]. TPU properties give footwear manufacturers the freedom to create unique designs [24,25]. TPU can also meet current requirements such as anti-static properties, increased resistance to abrasion, flexibility at low temperatures and anti-microbial activity, and the most important property is the one relating to comfort. These properties can be improved by using compatibilizing agents (polyethylene-graft-maleic anhydride—PE-g-MA) and reinforcement agents (with natural fibres modified with polydimethylsiloxane) [26,27]. PE-g-MA improves the properties of polymeric biocomposites such as tensile strength, tear strength, resistance to chemical agents, etc. The modification of leather (protein) and elastomeric waste (SBR rubber) in a mixture with polydimethylsiloxane (PDMS) was carried out in order to activate it, at the same time being a method of improving the wetting capacity, but also of binding through chemical interactions with the polymer matrix used to obtain the desired composite. At the same time, polydimethylsiloxane also acts as a plasticizer in the mixture [18,28,29,30,31,32].

In this work, biocomposites based on TPU, recycled thermoplastic polyurethane waste, protein (leather) and elastomeric waste (butadiene-styrene rubber—SBR) were obtained in a mixture, modified with PDMS and compatibilized with PE-g-MA. The biocomposites were characterized in terms of physico-mechanical properties (normal state, accelerated aging and atmospheric conditions and exposure for 365 days) and flow indices, and they were also subjected to FT-IR structural characterization and FT-IR microscopy, as well as differential scanning calorimetry—DSC—to determine thermal behaviour.

## 2. Experimental Section

### 2.1. Materials

Materials used in obtaining polymeric biocomposites were as follows:Thermoplastic polyurethane (TPU), from MD Graphene SL, Spain, is used in the footwear industry due to its thermoplasticity (it can be cast and injected into forming moulds), and PU—polyurethanes—used mainly in the footwear industry, are the product of a chemical reaction between a mixture of resins based on processed polyols [polyol resin-blend] and an aliphatic or aromatic isocyanate to obtain a micro-cellular structure. The reaction of “-OH” groups from polyols, polyesters or polyethers with “-N=C=O” groups from isocyanates leads to the formation of urethanes [24]. TPU has specific gravity (1.03 g/cc), hardness (65–85 Sh°A), tensile strength (>20 N/mm^2^), colour (black) and melt temperature (between 170° and 190 °C).Polyethylene grafted with maleic anhydride (PE-g-AM) from PolyRam Group, Israel is used as compatibilizer. PE-g-MA has the role of reducing the interfacial tension, achieving a fine dispersion of the ingredients, providing adhesion between phases in the solid state and at the same time stabilizing the morphology of the thermal effects during processing and also improving the physical–mechanical properties: tensile and tear strength, resistance to chemical agents, etc. The PE-g-MA compatibilizer has the following properties: density—0.91 g/cm^3^; hardness—45 Sh°D; melting point—117 °C; MFI—2 g/10 min (190 °C/2.16 kg); viscosity 330,000 cps; colour—honey yellow.Recycled thermoplastic polyurethane waste (TPUW) comes from the footwear industry. TPUW is cryogenically ground to sizes of approximately 0.5 mm.Mixed leather and SBR rubber (butadiene-styrene) waste from the footwear industry is cryogenically ground to micrometric sizes. The mixed leather and SBR rubber waste is used as a filling material, but at the same time, it has the role of a reinforcing agent. The incorporation of reinforcing agents in the polymer matrix improves the physical–mechanical properties of the obtained products.Polydimethylsiloxane fluid (PDMS), from Sigma-Aldrich, Inc., St. Louis, MO, USA. PDMS has the role of a plasticizer, but at the same time, it improves the dispersion of mixed protein and rubber waste in the polymer matrix.

### 2.2. Preparation of Biopolymeric Composites Based on TPU/TPUW/Protein and Elastomeric Waste in Mixture/Compatibilizer

Polymeric biocomposites based on thermoplastic polyurethane (TPU), recycled thermoplastic polyurethane waste (TPUW), mixed protein (leather) and elastomeric waste (SBR rubber—butadiene styrene) unmodified/modified with PDSM (polydimethylsiloxane) and polyethylene-graft-maleic anhydride (PE-g-MA) as compatibilizer were obtained by the mixing technique. The name of the samples and the composition of the polymeric biocomposites based on TPU/TPUW/mixed protein and elastomeric waste/compatibilizer are presented in Table 1 (percentages). The procedure for obtaining polymeric biocomposites includes the following stages (Figure 1):Collection of mixed protein and SBR rubber waste (from the footwear industry) and recycled TPU waste;Grinding of the mixed protein and elastomeric waste (SBR rubber) to micrometric dimensions of 0.35 mm, with a cryogenic cyclone mill (Retsch ZM 200, Verder Scien-tific, Haan, Germany) at a speed of 12,000 rpm, using dry ice as a cooling agent;The recycled TPUW waste is cryogenically ground to dimensions of approximately 0.5 mm using a cryogenic mill (Retsch ZM 200, Verder Scientific, Germany), at a speed of 10,000 rpm. Dry ice in the form of flakes is used as a cooling agent.Dosage of raw materials conducted was made according to the recipe in Table 1;Making the polymeric biocomposite in a Plasti-Corder Brabender Mixer 350E with a capacity of 350 cm^3^ (Brabender GmbH & Co. KG, Duinsburg, Germany); working temperature set at 160 °C. TPU is introduced for plasticization for 2′ at 30 rpm. After its plasticization, recycled TPUW waste is added (in the proportion of 20%, 60% and 80%) and mixed leather and SBR rubber waste unmodified/modified with PDMS, strictly following the order of introduction of the ingredients (Table 1) for 4′, at 30 rpm. Continue mixing until the mixture is homogenized for 5′, at a temperature of 160 °C at 80 rpm;Rheological testing to determine the flow indices, melt flow index (MFI—Haake Meltfix 2000, Haake Technic GMBh, Vreden, Germany), was carried out at a temperature of 190 °C with a pressing force of 5 kg, preheating for 4 min;Obtaining standardized plates by pressing in forming moulds on a laboratory hydraulic press (Fortune Press, model TP/600, Fontijine Grotness, Vlaardingen, The Netherland) via the method of compression between its plates at a temperature of 170 °C, preheating for 3 min, pressing for 3 min, and cooling with water for 10 min at a pressure of 300 kN. Samples with a size of 150 × 150 × 2 mm are obtained, which are left to condition for 24 h at room temperature for testing;Physical–mechanical tests, rheological tests (MFI—melt flow index), FT-IR structural characterization (Nicolet, Waltham, MA, USA) and FT-IR microscopy (Nicolet, Waltham, MA, USA), but also determination of thermal behaviour by thermogravimetry (TG) and differential scanning calorimetry—DSC (Netzsch 449C F3 STA Jupiter, Selb, Germany).

### 2.3. Modification of the Protein and Elastomeric Waste in Mixture

The mixed protein (leather) and SBR rubber waste (from the footwear industry) was modified by cryogenic grinding. Grinding was carried out using a cryogenic cyclone mill, with dimensions of 0.35 mm, at a speed of 12,000 rpm, and dry ice in the form of 3 cm pellets was used as a cooling agent. Modification of the mixed protein and elastomeric waste was achieved by contacting 100 g of post-consumer waste with 5% PDMS, under continuous mixing at 70 rpm for 2 h. The obtained mixture was then kept at a temperature of 70 °C, in an oven with circulating air, for 4 h, homogenizing the sample every 15–20 min.

### 2.4. Characterization of Biopolymeric Composite

The polymeric biocomposites were characterized from a physical–mechanical point of view [33]—hardness, elasticity, tensile strength, elongation at break (normal state at ambient temperature, accelerated aging at 70 °C for 168 h and at atmospheric conditions and weather for 365 days)—according to the standards in force, and the fluidity index (MFI—melt flow index) with the help of the Haake Meltfix 2000 equipment.

The physical–mechanical characterizations were carried out by taking samples from the plates obtained on the electric press at the established parameters, by stamping with standardized devices—punch knives (Figure 2).

Hardness of polymeric biocomposites based on TPU/TPUW/mixed leather and SBR rubber waste/PE-g-MA was measured in Shore°A and determined according to ISO 48-4:2018 and ASTM D2240 [34,36]. Hardness was determined on samples with flat surfaces with thickness of minimum 6 mm (minimum 3 readings, the result being the average of obtained values). Elasticity (%) was determined according to ISO 4662:2017 and ASTM D78121-05 [34,37], using dumbbell-shaped samples (3–4 samples are used) with a thickness of 2 ± 0.3 mm, and measurements were performed using a Schob test instrument. The calculation was according to Equation (1) [34,35].
E = [ (L_r_ − L_0_)/L_0_] × 100 [%],(1)
where L_r_ is the distance between the reference lines of the marking at the time of breaking, mm, and L_0_ is the initial distance between the reference lines of the marking, mm.

Tensile strength (N/mm^2^) and elongation at break (N/mm^2^) were determined according to ISO 37-2020 and ASTM D41 [34,37] standards on dumbbell-type specimens with a thickness of 2 ± 0.3 mm, with a Schopper dynamometer at a testing speed of 500 mm/min, taking into account the average of three determinations, Equation (2) [35,38],
R = F/(2 × S_0_) N/mm^2^,(2)
where F is the breaking force, N, and S_0_ represents the initial surface of the straight section, mm^2^.

Resistance to accelerated aging [35] is determined under heat conditions, via the hot air circulation oven method at 70 ± 1 °C and 168 h and at atmospheric pressure, temperature and time (365 days of atmospheric conditions: rain, wind, hail, sun); tests were performed according to ISO 188/2011 [39]. Before the samples are subjected to physical–mechanical determinations, they are conditioned for 16 h. Resistance to atmospheric pressure, temperature and time (365 days of atmospheric conditions: rain, wind, hail, sun) is also carried out on dumbbell-type samples (3 readings) according to the SR ISO 188/2011, and the results are compared with those obtained under normal conditions. By determining the two characteristics, changes in the appearance of the surface of the samples such as cracks and colour changes can be observed, and they can be harder or softer, implicit in the physical–mechanical characteristics.

The fluidity index (MFI) is determined according to ISO 1133/2012 [34,40], and the relationship that defines the flow index is (Equation (3)) [35]:MFR (θ, m_nom_) = (t_ref_ × m)/t,(3)
where MFR is the flow index, θ is the test temperature, in °C, m_nom_ is the nominal load, in kg, t_ref_ is reference time, in minutes (10 min), m is the average mass of the extruded product, in grams (g), and t is the time interval between two cuts of the extruded product, in seconds (s).

Thermal behaviour was determined by thermal analysis TG-DSC using a Netzsch 449C F3 STA Jupiter device. Samples were placed in a closed aluminium crucible and heated at 10 K min^−1^ from room temperature up to 900 °C, in dry air atmosphere, at a flow rate of 50 mL min^−1^. The evolved gases were transferred trough heated transfer lines and analysed on the fly with help of a FTIR Tensor 27 from Bruker (Bruker Co., Ettlingen, Germany), equipped with an internal thermostatic gas cell.

Fourier Transform Infrared Spectroscopy (FTIR) of samples was obtained using Nicolet iS50 FT-IR spectrophotometer with ATR and diamond crystal, in the wave number ranging from 4000 cm^−1^ to 400 cm^−1^. The FTIR 2D maps were recorded with a Nicolet iN10 MX in the domain 4000–650 cm^−1^.

## 3. Results and Discussion

### 3.1. Melt Flow Index Determination

The fluidity indices were determined at a temperature of 190 °C, with a pressing force of 5 kg, preheating for 4 min. The more viscous the materials, the more pressing force they require to be extruded through the MFI die [40].

According to Table 2 and Table 3, it can be observed that the flow index of the polymeric biocomposites based on TPU, recycled TPUW waste and mixed leather and SBR rubber waste (styrene butadiene) are influenced by the working temperature, the amount of recycled TPUW waste introduced into the mix and the presence of PE-g-MA compatibilizer. The flow index values, Table 3, decrease significantly proportionally, in the case of formulations compatibilized with PE-g-MA, with the amount of mixed leather and SBR rubber waste compared to the reference sample (control)—MM, due to the increase in melt viscosity. The flow index values of polymeric biocomposite formulations based on TPU and recycled TPUW waste, Table 2, increase compared to the control sample—MM. Thus, the technological process is controlled by the correlation that exists between the basic properties of the new polymer structures obtained and the parameters of the technological process. When flow index values decrease, the flow properties and surface appearance of materials change significantly. Thus, polymeric biocomposites with high flow indices such as T60 and T80 can be used for injection processing, and those with low flow indices such as TBB1 and TBB2, but also the TBB11–TBB13 series, can be used for processing in the press in forming moulds. We can thus appreciate that biocomposites compatibilized with PE-g-MA can be used in the footwear industry.

### 3.2. Physical–Mechanical Characterisation of Polymeric Biocomposites Based on TPU/TPUW/Mixed Leather and SBR Rubber Waste/PE-g-MA

For the physico-mechanical characterization (normal state, accelerated aging and atmospheric and weather conditions for 365 days), 15 × 15 cm plates were obtained on the laboratory electric press between its plates at well-established specific parameters, Table 4. The following mechanical characteristics were determined: hardness (Sh°A), elasticity (%), tensile strength (N/mm^2^) and elongation to break (N/mm^2^), according to the standard in force (Section 2.4) [24,25,26,41,42]. All results in Table 4 are shown as mean values (minimum 3 determinations according to the standards in force) ± standard deviation (SD).

For polymeric composites based on TPU compounded with recycled TPUW waste in proportions of 20, 60 and 80%% (recycled waste) in normal state and accelerated aging, it is observed that the hardness in normal state at ambient temperature is influenced by recycled TPUW waste added to the mix [11]. As the percentage of added waste increases, the hardness decreases by 3–6 Sh°A, and after the accelerated aging process at 70 °C for 168 h and atmospheric conditions for 1 year, hardness increases by 1–6 Sh°A. The introduction of the compatibilizing agent PE-g-MA in a percentage of 5%, but also the modification of the protein waste and SBR rubber mixed with 5% PDMS (polydimethylsiloxane), changes the mechanical properties of the polymeric biocomposites [43]. PDMS has the role of a plasticizer and also improves the dispersion of waste in the polymer matrix. [18,18,20,28,44]. The hardness for samples TBB1 and TBB2 increases by 5–6 Sh°A. For the TBB11-TBB13 series, by introducing the compatibilizing agent, the hardness values increase significantly by approximately 9 Sh°A. After the process of accelerated aging and atmospheric conditions for 1 year, the hardness values increase significantly.

The elasticity values decrease by up to 14.28%, in normal condition (T20–T80). For the same series of samples, T20–T80, after the accelerated aging process at 70 °C for 168 h and at atmospheric and weather conditions for 365 days, the elasticity shows a slight decrease in the calculated values, by 8.33%, and by 4.54, respectively, to 13.63%. The elasticity in normal condition, due to the compatibilization with PE-g-MA (TBB11, TBB12, TBB13), decreases significantly by 21.4% compared to the control sample MM. After the samples were subjected to the accelerated aging process and atmospheric conditions for 1 year, we can observe a decrease in elasticity by 8.33% (TBB1 and TBB2), and by values between 9.09% and 13.66%, respectively, for the TBB11-TBB1 series, compared to the control sample—MM [11,20].

The tensile strength in normal condition, T20–T60, increases by 16.7 up to 22.23%, compared to the control, having values from 7.4 N/mm^2^ to 7.75 N/mm^2^. After being subjected to accelerated aging and atmospheric conditions (weathering) for 365 days, it presents values between 7.74 N/mm^2^ and 7.91 N/mm^2^, the tensile strength increasing by 1.97 to 4.2%. In the case of samples with recycled TPUW waste (TBB11), mixed leather and SBR rubber waste modified with 5% PDMS and unmodified (TBB12 and TBB13) and compatibilized with PE-g-MA (5%), the tensile strength shows an increase of 34.22%, and after the accelerated aging process at 70 °C × 168 h, the tensile strength decreases by up to 6.58% [11,18,20].

The elongation at break for the T20–T80 series increases, both in normal condition and accelerated aging, but also under atmospheric conditions for 1 year. For T20, T60 and T80 fabrics (samples containing recycled TPU/TPUW) the elongation at break increases with values in the range 27.27–26.36%. The determinations were carried out under normal conditions after the samples were conditioned for 24 h at room temperature according to the standards in force [11,20]. The elongation at break for the TBB11-TBB13 series, both in normal condition and accelerated aging, decreases depending on the type of waste, but also on the PE-g-MA compatibilizer added to the mixture.

According to the standard for use in the footwear industry, samples T60 (TPU/TPUW 60%) and TBB12 (TPU/unmodified leather and SBR rubber waste/PE-g-MA) present optimal values that are suitable for use in the footwear industry.

### 3.3. FT-IR Spectroscopy

Polymeric biocomposites based on TPU, TPUW waste, mixed leather and SBR rubber waste with PE-g-MA were characterized in terms of FT-IR spectroscopy, and the determination was performed in the wave number ranging from 4000 cm^−1^ to 400 cm^−1^ [45,46].

The FT-IR spectrum of thermoplastic polyurethane (TPU) is shown in Figure 3.

The spectrum obtained for the control sample MM (TPU), Figure 3, highlights the characteristic adsorption bands originating from the polyurethane functional groups. Thus, the band at 3327.28 cm^−1^ is associated with the stretching vibration of N-H bonds, which indicates the presence of hydrogen bonds between the NH groups and the macro diol ester groups (C=O). The peaks appearing in the range 2955.37–2851.11 cm^−1^ correspond to the asymmetric and symmetric stretching vibrations of the aliphatic CH_2_ groups present in the TPU structure. The presence of C-H bonds is also confirmed in the region 1451.43–1413.65 cm^−1^. The intense adsorption band at 1726.28 cm^−1^ comes from the stretching vibration of C=O bonds (known as Amide I). Polyurethanes are able to form more hydrogen bonds due to the presence of an N-H donor group and a C=O acceptor group in the urethane bond. The characteristic peak originating from polyurethane at 1529.66 cm^−1^ indicates the stretching vibration of C-N bonds (Amide II). The band at 1596.42 cm^−1^ represents the stretching vibration (in the plane) of the C=C bonds of the aromatic rings that come from the diisocyanate units. Bands in the range 1220.34–1253.86 cm^−1^ (Amide III) indicate stretching vibration of CN bonds and bending of NH bonds; the band in the range 1140–1120 cm^−1^ represents the symmetric stretching vibration of the COC bond (confirms the urethane bond); the peak at 1310.2 cm^−1^ corresponds to the C-O group present in esters, and the one at 1069.86 cm^−1^ comes from the stretching vibration of C-O-C bonds [46]. The band at 816.23 cm^−1^ comes from the C-H bending vibration of the p-disubstituted benzene ring [47,48,49,50,51].

In the case of mixtures containing varying percentages of thermoplastic polyurethane waste—T20, T60 and T80 (Figure 4)—the spectra are similar to the ones obtained for the MM sample, which demonstrates that repeated reprocessing at the temperatures at which the experiments took place did not lead to thermal degradation/oxidation processes [46,47,48,49,50,51]. This is due to the fact that reprocessing at the temperatures at which the experiments were carried out at 160 °C do not induce thermal degradation or oxidation processes, a fact also confirmed by the thermal analysis, which shows the very good stability of these mixtures up to a temperature of 240 °C. Moreover, it is known that the TPU block-copolymer chain is composed both of a hard isocyanate segment and of soft segments based on esters/ethers. According to specialized studies, the degradation processes of the hard segments begin at approximately 200 °C, while for the soft segments, the degradation begins at temperatures higher than 300 °C. Therefore, the temperatures at which the experiments took place, approximately 160 °C, but also the relatively short processing time (11 min), are considered safe and ensure the optimal processability of such mixtures [52].

Figure 5 shows the FTIR spectra obtained both for the unmodified mixed leather and SBR rubber waste, and for the waste mixture modified with PDMS. The characteristic bands originating from SBR rubber can be visualized at 2918.07 and 2849.07 cm^−1^ (attributed to the stretching vibration of the -CH groups from the styrene aromatic ring), the band at 964.94 cm^−1^ is attributed to 1,4 trans butadiene units, and the one at 910.48 cm^−1^ is associated with units originating from the 1, 2 butadiene bond. The peak at 698.75 cm^−1^ comes from the out-of-plane deformation of the C=C bond of the polystyrene benzene ring. The band found at 1450.74 cm^−1^ represents the in-plane deformation of -CH_2_ bond, and the one at 1491.77 cm^−1^ is associated with C=C bond originating from the aromatic ring. The bands originating from the protein waste can be visualized at 3298.31 cm^−1^ (associated with N-H stretching bond) and at 1639.31 cm^−1^ (Amide I—stretching vibration of C=O bond originating from the protein structure). These specific protein bands can be identified only in samples containing leather wasted (and they are of course missing in the simple SBR sample). The band at 1538.94 cm^−1^ is known as Amide II and can be associated with the bending vibration of N-H bond and the stretching vibration of C-H bond. In addition to the bands originating from SBR rubber and protein fibres, the bands originating from SiO_2_/kaolin can also be visualized, identified on the basis of its intense characteristic bands at 1084.97 cm^−1^ and 459.45 cm^−1^ (bands originating from the Si-O-Si bond, Si-O). If the mixed leather and SBR rubber waste is modified with PDMS, the appearance of new bands can be visualized, especially at 1259.36 cm^−1^ (the symmetric bending vibration of CH_3_ bond originating from the Si-CH_3_ group) and 798.71 cm^−1^ (rocking vibration of CH_3_ bond in the Si-CH_3_ bond) [28,53].

In Figure 6, the spectrum obtained for sample TBB1, a mixture based on virgin TPU compounded with mixed leather and SBR rubber waste, and for TBB2, virgin TPU compounded with mixed leather and SBR rubber waste modified with 5% PDMS, the characteristic bands of TPU can be highlighted at ~1726, 1596, 1310 cm^−1^, etc. But, the adsorption band at 1639 cm^−1^ (associated with C=O bonds or the Amide I band) originating from the protein waste has a very low intensity in the case of TBB1 and TBB2 mixtures. This is due to the fact that the amount of protein waste and SBR rubber mixture is only 20% in these samples, and therefore, these wastes are practically embedded in the TPU matrix, which screens the signal, and therefore, the adsorption band from ~1639 cm^−1^ cannot be detected in ATR. This decreasing intensity is further highlighted by the fact that C-H vibration originating from the aromatic part of the styrene, 3000–3100 cm^−1^, visible in the protein waste and SBR rubber mixture, cannot be observed in the TBB 1 and TBB2 samples.

For sample TBB11 (Figure 7A), mixture based on virgin TPU/recycled TPU/5% PE-g-AM, the intense bands originating from TPU can be visualized at 3327.28, 1726.28, 1529.66 and 1413.65 cm^−1^, respectively. The presence of PE-g-MA is difficult to detect, because the characteristic bands originating from it overlap with those of TPU, especially the C=O band, from approximately 1726 cm^−1^, which can be attributed to the asymmetric and symmetric stretching vibrations which come from maleic anhydride and the carboxylic group from maleic acid. Other bands originating from PE-g-AM, such as the one at 2919 cm^−1^_,_ correspond to the asymmetric stretching vibrations of the -C-H bonds originating from ethylene units and is common with TPU. Other bands overlaid with TPU that may originate from PE-g-MA are at approximately 1450 cm^−1^ and 713 cm^−1^ and represent the bending and rocking vibrations of aliphatic bonds originating from PE-g-MA [54,55]. For a better highlighting of the overlap of the bands originating from PE-g-AM with those originating from TPU, see Figure 7B. As can be seen, most of the bands are common, due to similar bonds type. In addition, from the spectra, the presence of PE-g-AM cannot be detected in the case of the TBB11 mixture (having the composition 80% TPU/20% recycled TPU/5% PE-g-AM) compared to the T20 mixture considered in this case as the control sample (composed of 80%/TPU/20% recycled TPU). This can be attributed on the one hand to the low concentration of grafted maleic anhydride on the PE surface (see the small band at about 1715.92 cm^−1^, observed in the pure spectrum of PE-g-AM originating from the C=O bond of the maleic anhydride group), but also of its low concentration, of only 5% added in the TBB11 mixture. These results are in agreement with the observations made by other authors regarding the impossibility of detecting PE-g-AM in different mixtures [56,57].

In the case of samples containing waste in a mixture of leather and SBR rubber in a TPU matrix, samples TBB12 and TBB13, the presence of these wastes can be visualized at approximately 963 cm^−1^ and 468 cm^−1^, respectively. The intensity of these bands is higher in sample TBB13, which contains waste functionalized with 5% PDMS.

### 3.4. FT-IR Mapping Investigation

The FTIR 2D maps for the polymeric biocomposite based on TPU/TPUW/mixed protein and SBR rubber waste unmodified/modified with 5%PDMS/PE-g-MA were recorded in the domain 4000–650 cm^−1^ in order to determine homogeneity and inclusion of recycled TPUW waste in the mixtures, as well as of mixed leather and SBR rubber unmodified or modified with PDMS [58,59]. FT-IR mapping investigation was made at 3308 cm^−1^, 2926 cm^−1^ and 1727 cm^−1^ and FT-IR microscopy at selected wavelengths. Red areas indicate the highest absorbance, while blue areas correspond to the lowest absorbance.

For the control sample MM (virgin TPU), based on the FT-IR analysis and the recorded spectrum, Figure 8a, the characteristic bands coming from the functional groups of the polyurethane in the range 3000–2800 cm^−1^, as well as 1800–1700 cm^−1^, stand out, due to the characteristic peak. The peak that appears in the area of 1800–1700 cm^−1^ comes from the stretching vibration of the C=O bond, also known as Amide I. From the FT-IR microscopy of the TPU sample, we can see an almost undisturbed surface of the sample, indicating that plasticization took place according to the working parameters (temperature, plasticization time and homogenization).

In the case of the sample containing TPU/TPUW, Figure 8b, sample T60, we can observe in the recorded FT-IR spectra a good homogenization of the recycled TPUW waste in the mass of virgin TPU without degradation of the obtained compound, even though the waste is subject to repeated reprocessing (up to a maximum of five reprocessing cycles) [59].

In Figure 8c–e, showing biocomposites based on TPU/TPUW/mixed leather and SBR rubber waste unmodified/modified with PDMS/PE-g-MA, for samples TBB11, TBB12 and TBB13, we can observe the stretching bands coming from virgin thermoplastic polyurethane in the range 1700–1800 cm^−1^, and the peaks in the range 1400–1600 cm^−1^, respectively. From the recorded FT-IR spectra, we observe the best absorbance in the case of sample TBB12 compared to samples TBB11 and TBB13. The recorded micrograph shows a good compounding (homogenization) of the mixed protein and SBR rubber waste, and the PE-g-MA compatibilizer added in a proportion of 5%, favouring a good dispersion of the elastomer in the polymeric biocomposite composition obtained.

In the case of sample TBB11, which contains virgin TPU (80%), recycled TPU waste (TPUW) (20%) and 5% PE-g-MA (Figure 8c), a slight agglomeration is observed. This is due to the fact that this sample does not contain waste modified with PDMS, which helps to eliminate the tendency of waste particles to agglomerate. At the same time, this procedure ensures a better dispersion of the waste particles in the polymer matrix. PDMS also improves the role of plasticizer at the same time and the dispersion of waste in the polymer matrix. The impact on the mechanical properties of the biocomposite are not very different compared to the control sample MM (virgin TPU) and sample T20 (without PE-g-MA). These characteristics are slightly influenced due to the use of 5% PE-g-MA compatibilizer, which improves the properties of the obtained biocomposites.

### 3.5. Thermal Analysis TG-DSC

The polymeric biocomposites (MM—control, T60, TBB11, TBB12, TBB13) were analysed from the point of view of thermal behaviour by thermogravimetry and differential scanning calorimetry coupled with FTIR analysis of evolved gases (Figure 9 and Figure 10) [60,61,62,63].

All samples exhibit a good stability up to 240 °C with less than 3% mass loss. The MM sample lost 1.48% of its initial mass up to 240 °C due to elimination of adsorbed water molecules as indicated by the FTIR of evolved gases. The onset melting temperature (T_on_) was determined as 181.5 °C, with the peak at 195.3 °C corresponding to the end of the melting process. In the temperature interval 240–425 °C, the sample suffers an oxidative degradative process, with a recorded mass loss of 67.75%. The FTIR of evolved gases indicates the presence of high quantities of CO_2_ (2354 cm^−1^) and some H_2_O (3500–3900 cm^−1^) resulting from oxidation reactions, but also an important amount of hydrocarbons, both aliphatic and aromatic, resulting in the polymer backbone fragmentation (wavenumbers around 3000 cm^−1^). After 435 °C, the residual carbonaceous mass is burned away, with the recorded mass loss being 24.83% up to 900 °C. The evolved gases contain mainly CO_2_ in this temperature interval. The residual mass is 6.15%. The most important information from thermal analysis is presented in Table 5.

The data from Table 5 indicate that the sample T60 has the lower value for the melting peak, at 188.9 °C, while the sample TBB12 presents the highest value, at 199.8 °C. The addition of TPUW, which most probably has shorter polymer chains and is less reticulated, induced specific modification of properties. The differences observable in thermal analysis, induced by the different composition of the samples, indicate that the presence of recycled TPU in the samples T60 and TBB11 leads to less fragmentation of polymeric backbone and promotes the oxidation reactions, with the FTIR diagrams (Figure 10) indicating the presence of small amounts of hydrocarbons along mainly CO_2_. The fact that these two samples are rather oxidized than decomposed can also be seen from the strong exothermic effect from 500–600 °C (Figure 9). Of these two samples, TBB11 exhibits a slightly higher thermal stability, as can be observed from both TG and DSC curves.

## 4. Conclusions

In a growing society, people are aware of the importance of the environment. Recycle reduces stress on the environment. Thus, biocomposites are developed to be recycled by the simplest method. In the first phase, the recycling is carried out by coarse grinding. The reuse of the material is in the form of granules of different sizes. This cycle of recycling and reuse can be repeated up to a maximum of five times, without significant degradation of the properties, thus reducing the impact on the environment and the carbon footprint, by closing the loop and extending the period of use of the product.

One of the main advantages of polymeric biocomposites based on TPU, recycled TPU waste and mixed leather and SBR rubber waste unmodified/modified with PDMS and compatibilized with PE-g-MA is that the use of waste from the footwear industry reduces pollution. Thus, we protect the environment by transforming waste through different processing methods into new products with added value, and at the same time, we protect the human factor by reducing the toxicity of the work environment. The polymeric biocomposites were made on devices specific to elastomers and plastics, and they were tested according to the standards in force. The modification of the leather and SBR rubber waste with PDMS was carried out in order to activate it, at the same time having the role of a plasticizer in the mixture. The PE-g-MA compatibilizer improves physical–mechanical properties: tensile strength, tear strength, etc. Thus, by increasing the percentage of recycled TPUW waste, but also by adding protein waste and SBR rubber unmodified/modified with PDMS (which also acts as a reinforcing agent), physical–mechanical properties such as hardness and tear strength increase. With regard to rheological analysis, the MFI indicates that the polymeric biocomposites were made at optimal working parameters.

Following the tests carried out, according to the standard for use in the footwear industry, samples T60 (TPU/TPUW 60%) and TBB12 (TPU/unmodified leather and SBR rubber waste/PE-g-AM) present optimal values that are suitable for use in the footwear industry. T60 can be used for injection moulding and TBB12 for press moulding in forming moulds.

## Figures and Tables

**Figure 1 materials-16-05279-f001:**
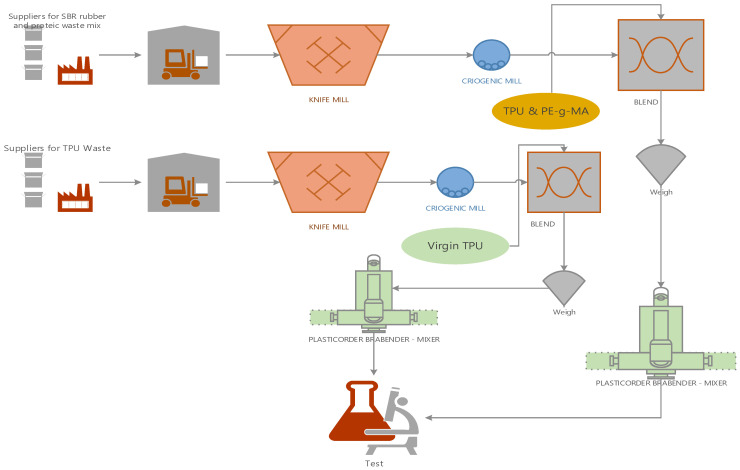
Stages of obtaining the polymeric biocomposite based on TPU/TPUW/mixed leather and SBR rubber waste/PE-g-MA.

**Figure 2 materials-16-05279-f002:**
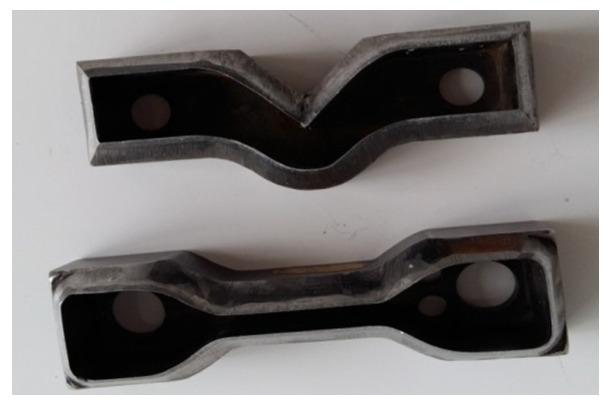
Punch knives for polymeric biocomposite samples based on TPU/TPUW/mixed leather and SBR rubber waste/PE-g-MA [34,35].

**Figure 3 materials-16-05279-f003:**
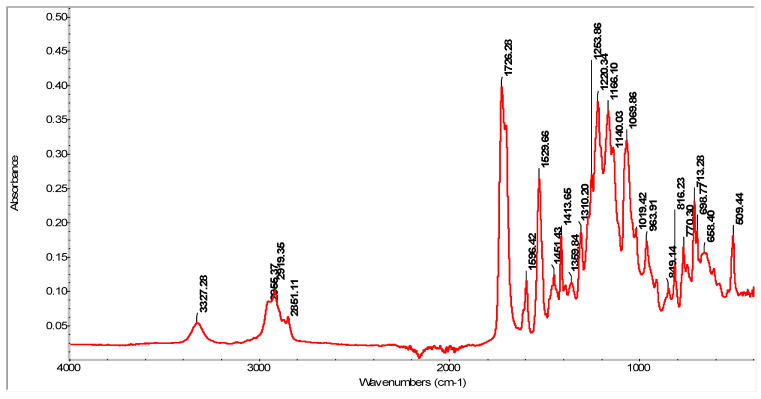
FT-IR spectrum of virgin thermoplastic polyurethane—control sample, MM.

**Figure 4 materials-16-05279-f004:**
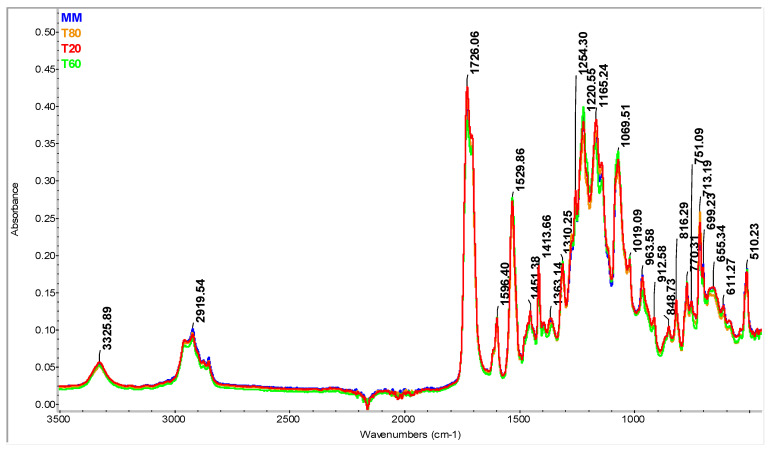
FT-IR spectrum of virgin TPU samples (MM) with varying percentages of 20, 60 and 80% TPUW waste (T20, T60, T80).

**Figure 5 materials-16-05279-f005:**
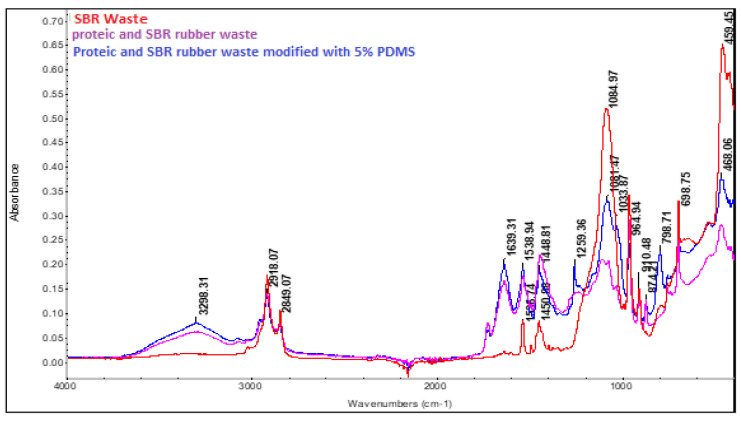
FTIR spectrum of mixed leather and SBR rubber waste, and mixed leather and SBR rubber waste modified with 5% PDMS, respectively.

**Figure 6 materials-16-05279-f006:**
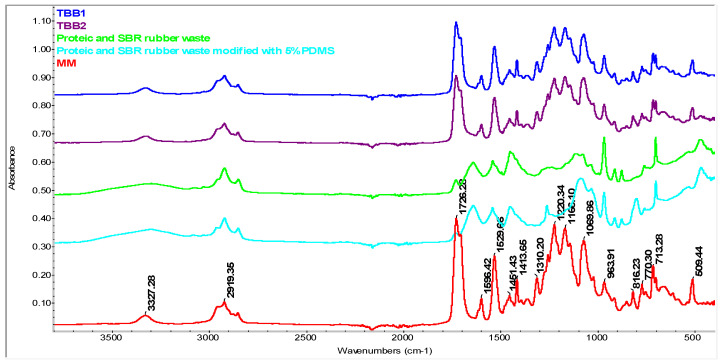
FTIR spectrum of biocomposites based on TPU compounded with mixed leather and SBR rubber waste (TBB1), and mixed leather and SBR rubber waste modified with 5% PDMS (TBB2), respectively.

**Figure 7 materials-16-05279-f007:**
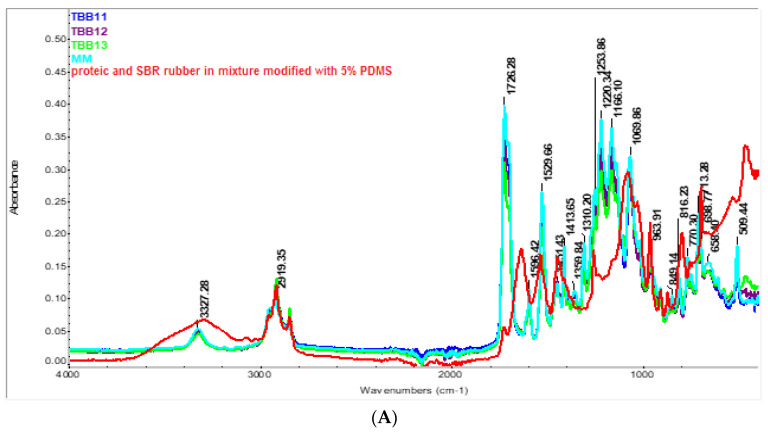
FTIR spectrum of biocomposites based on TPU compounded with recycled TPUW, mixed leather and SBR rubber waste unmodified/modified with PDMS (**A**) FTIR spectrum of protein and SBR rubber in mixture with 5% PDMS and MM, TBB11, TBB12, TBB13; (**B**) FTIR spectrum of PE-g-MA and T20 and TBB11 samples.

**Figure 8 materials-16-05279-f008:**
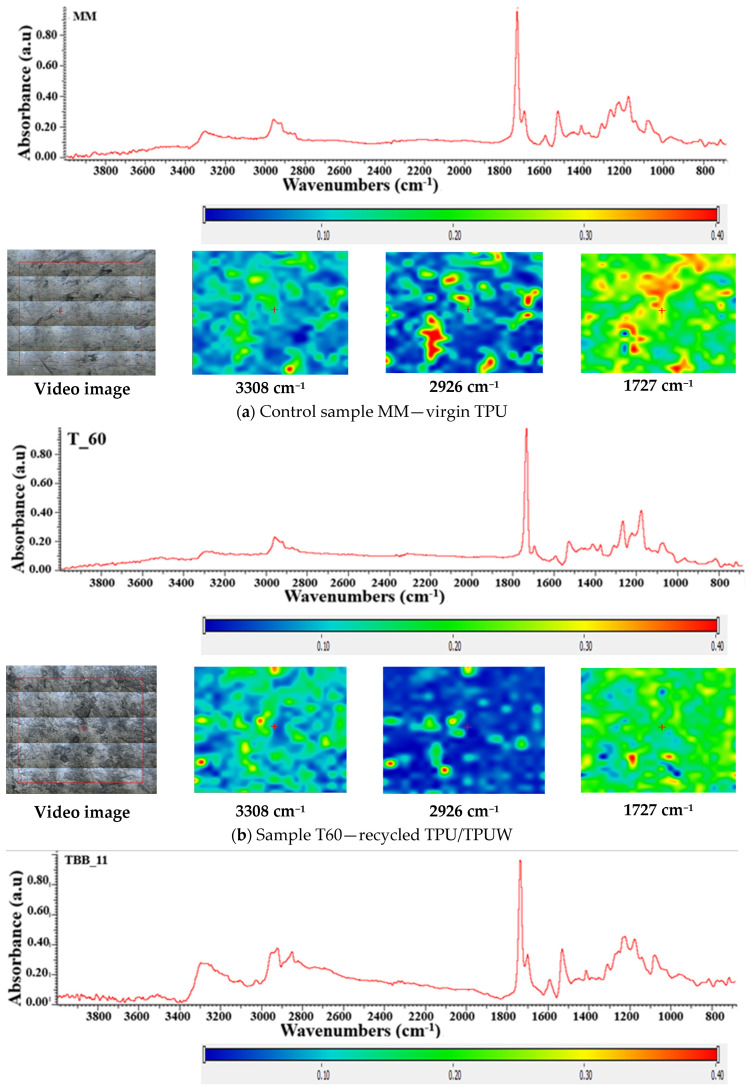
FT-IR mapping investigation at 3308 cm^−1^, 2926 cm^−1^ and 1727 cm^−1^ of biocomposites based on TPU/TPUW/mixed leather and SBR rubber waste unmodified/modified with PDMS/PE-g-MA (MM—control, T60, TBB11, TBB11, TBB13)—(**a**–**e**).

**Figure 9 materials-16-05279-f009:**
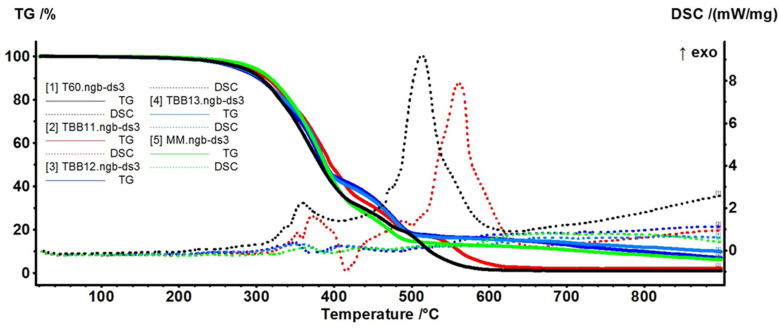
TG-DSC curves for the biocomposites MM—control, T60, TBB11, TBB11, TBB13.

**Figure 10 materials-16-05279-f010:**
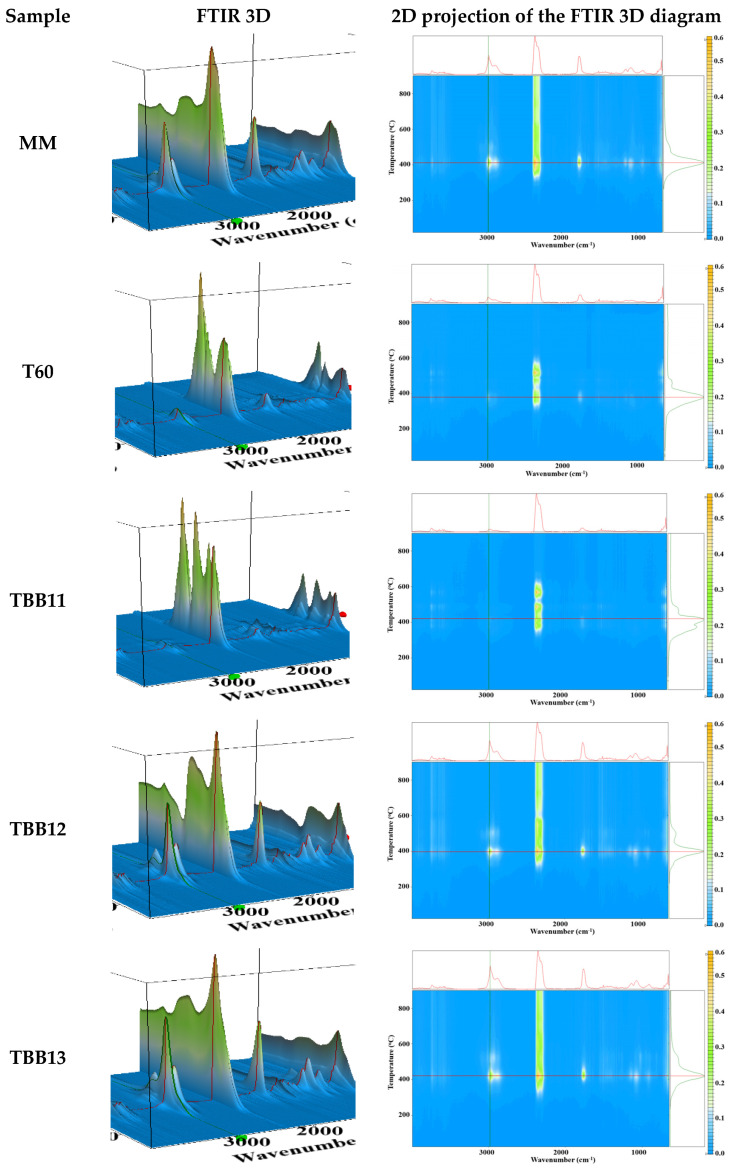
The FTIR 3D diagrams and their 2D projections in the temperature/wavenumber plane for the biocomposites MM—control, T60, TBB11, TBB11, TBB13. On top of each 2D projection is the FTIR spectrum at the temperature of highest decomposition rate; on the right side of each 2D projection is the evolving trace for the wavenumber 2972 cm^−1^ assigned to the C-H asymmetric vibration from –CH_3_ moieties.

**Table 1 materials-16-05279-t001:** Polymeric biocomposites based on TPU, TPUW waste, mixed leather and SBR waste and PE-g-MA (^a^).

Ingredients		Sample								
	UM	MM	T20	T60	T80	TBB1	TBB2	TBB11	TBB12	TBB13
TPU	%	100	80	40	20	80	80	80	80	80
Recycled TPU (TPUW)	%	0	20	60	80			20		
Leather and SBR rubber waste	%					20			20	
Leather and SBR rubber waste modified with 5% PDMS	%						20			20
PE-g-MA	%							5	5	5

^a^ Parts to 100 parts TPU (phr).

**Table 2 materials-16-05279-t002:** Values of flow indexes of polymeric biocomposites based on TPU/TPUW (the mean values ± SD—Standard deviation).

Working Temperature, °C		Sample			
	UM	MM	T20	T60	T80
190 °C, SR ISO 1133	g/10 min	43.3 ± 0.83	71.8 ± 0.998	166 ± 0.84	205 ± 0.47

**Table 3 materials-16-05279-t003:** Values of flow indexes of polymeric biocomposites based on TPU/TPUW/mixed leather and SBR rubber waste/PE-g-MA (the mean values ± SD—Standard deviation).

Working Temperature, °C		Sample					
	UM	MM	TBB1	TBB2	TBB11	TBB12	TBB13
190 °C, SR ISO 1133	g/10 min	43.3 ± 0.833	13.5 ± 0.6	6.59 ± 0.93	27.7 ± 0.72	16.7 ± 0.87	5.45 ± 0.91

**Table 4 materials-16-05279-t004:** Physical–mechanical characteristics of polymeric biocomposites based on TPU, TPUW waste, mixed leather and SBR rubber waste and PE-g-MA (the mean values ± SD—Standard deviation).

Physical–Mechanical Characteristics	Sample								
	MM (Control)	T20	T60	T80	TBB1	TBB2	TBB11	TBB12	TBB13
Normal state									
Hardness, Sh°A	83 ± 0.57	83 ± 0.57	80 ± 0.57	77 ± 0.57	88 ± 0.57	89 ± 0.57	83 ± 0.57	91 ± 0.57	91 ± 0.57
Elasticity, %	28 ± 0.14	26 ± 0.40	26 ± 0.40	24 ± 0.40	24 ± 0.40	24 ± 0.40	22 ± 0.40	22 ± 0.40	22 ± 0.40
Tensile strength, N/mm^2^	6.34 ± 0.69	7.75 ± 0.47	7.44 ± 0.34	7.4 ± 0.28	8.37 ± 0.41	9.34 ± 0.72	8.16 ± 0.45	8.1 ± 0.13	8.51 ± 0.16
Elongation at break, %	220 ± 20	280 ± 11.54	300 ± 0	300 ± 0	180 ± 11.54	100 ± 0	260 ± 11.54	180 ± 0	100 ± 0
Accelerated ageing at 70 °C, for 168 h									
Hardness, Sh°A	84 ± 0.57	85 ± 0.57	83 ± 0	82 ± 0.57	89 ± 0.57	90 ± 0.57	86 ± 0.57	92 ± 0.57	94 ± 0.57
Elasticity, %	24 ± 0.14	24 ± 0.14	24 ± 0	22 ± 0.4	22 ± 0.4	22 ± 0.4	22 ± 0.21	22 ± 0.2	22 ± 0.2
Tensile strength, N/mm^2^	7.59 ± 0.19	7.74 ± 0.75	7.8 ± 0.27	7.91 ± 0.23	7.97 ± 0.45	9.23 ± 0.13	7.09 ± 0.32	7.42 ± 0.23	7.15 ± 0.44
Elongation at break, %	260 ± 0	260 ± 20	320 ± 0	380 ± 20	200 ± 20	110 ± 10	250 ± 20	160 ± 10	100 ± 10
Atmospheric and weather conditions for 365 days									
Hardness, Sh°A	89 ± 0.57	89 ± 0.57	90 ± 0.57	81 ± 0.57	95 ± 0.57	96 ± 0.57	89 ± 0.57	95 ± 0	99 ± 0.57
Elasticity, %	22 ± 0.4	21 ± 0.2	21 ± 0.2	19 ± 0.28	19 ± 0.28	19 ± 0.28	20 ± 0.4	19 ± 0.23	19 ± 0.30
Tensile strength, N/mm^2^	5.72 ± 0.15	5.57 ± 0.15	5.60 ± 0.11	6.89 ± 0.12	6.97 ± 0.12	8.09 ± 0.22	6.28 ± 0.26	6.59 ± 0.20	5.95 ± 0.27
Elongation at break, %	290 ± 26	290 ± 26	360 ± 11.54	400 ± 11.54	240 ± 15.27	180 ± 11.54	180 ± 11.54	160 ± 10	140 ± 20

**Table 5 materials-16-05279-t005:** Principal data from thermal analysis of MM, T60, TBB11, TBB12 and TBB13 samples.

Sample	Mass Loss (%) RT-240 °C	T_on_ (°C) (PE-g-MA)	Melting Peak (°C)	T_on_ (°C) (TPU)	Melting Peak (°C)
MM	1.48%	-	-	181.5	195.3
T60	2.11%	-	-	180.9	188.9
TBB11	1.89%	91.8	107.3	177.1	192.3
TBB12	2.27%	93.1	106.2	187.4	199.8
TBB13	2.16%	92.5	105.4	179.3	195.7

## Data Availability

The data presented in this study are available upon request from the corresponding authors.

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
