# Peer review of "Polymeric Biocomposite Based on Thermoplastic Polyurethane (TPU) and Protein and Elastomeric Waste Mixture"

_materials, 2023, doi:10.3390/ma16155279_

Round 1

Reviewer 1 Report

Manuscript ID: materials-2488139

Title: Polymeric Biocomposite Based on Thermoplastic Polyurethane (TPU) and Protein and Elastomeric Waste Mixture

After conducting a thorough assessment, I have determined that significant revisions are necessary for the aforementioned manuscript.

1.       The FT-IR spectra obtained for the control sample MM (TPU) in Figure 3 show characteristic absorption bands associated with the polyurethane functional groups. Provide more details on the specific functional groups and their corresponding vibration modes. In Figure 4, the spectra of the mixtures containing varying percentages of TPU waste (T20, T60, T80) are similar to the control sample MM. explain why repeated reprocessing at the experimental temperatures did not lead to thermal degradation or oxidation processes?

2.       The FTIR spectra in Figure 5 display characteristic bands originating from SBR rubber, mixed leather, and protein waste. elaborate on the specific functional groups and their assignments in these spectra. In Figure 6, the spectra of samples TBB1 and TBB2 show characteristic bands of TPU along with a shoulder at approximately 1639 cm-1 associated with C=O bonds from the protein waste. explain the presence of this shoulder and its significance in the composition?

3.       The recorded FT-IR spectra for sample TBB11 in Figure 7 show intense bands originating from TPU. Discuss the presence and detectability of PE-g-MA in this sample and its overlapping bands with TPU. The FTIR 2D maps in Figure 8 provide insights into the homogeneity and inclusion of recycled TPUW waste, mixed leather, and SBR rubber waste in the biocomposites. Discuss the significance of the observed absorbance patterns and their correlation with the compound's properties?

4.       The thermal analysis results in Figure 9 and Table 5 indicate the stability and decomposition behavior of the polymeric biocomposites. explain the differences in melting peak temperatures and the presence of hydrocarbons and CO2 in the evolved gases. One of the main advantages mentioned is the reduction of pollution by utilizing waste from the footwear industry. Elaborate on the specific environmental benefits and the ways in which these polymeric biocomposites contribute to reducing toxicity in the work environment?

5.       It is mentioned that the modification of leather and SBR rubber waste with PDMS activates it and acts as a plasticizer. provide more details on the activation process and the role of PDMS as a plasticizer in the mixture. The addition of PE-g-MA compatibilizer is stated to improve physical-mechanical properties. discuss the specific mechanisms by which PE-g-MA enhances properties such as tensile strength and tear strength in the biocomposites?

6.       The mechanical properties of the polymeric biocomposites are reported in Table 6. It would be helpful to include the standard deviations alongside the mean values to provide a better understanding of the data variability and reliability.

7.       The SEM micrographs in Figure 10 depict the surface morphology of the biocomposites. provide a brief discussion on the observed differences in the microstructure between the samples and their correlation with the mechanical properties. The presence of agglomerates in the SEM images of TBB11 in Figure 10 is noticeable. elaborate on the possible reasons for this aggregation and its impact on the overall performance of the biocomposite?

8.       The water absorption capacity of the biocomposites is discussed in the text, but no experimental data or results are provided. It would be beneficial to include the quantitative measurements of water absorption for a comprehensive analysis.

9.       The recycling potential of the polymeric biocomposites is mentioned briefly in the conclusion. Expand on the recyclability aspects, such as the ease of recycling, potential degradation during recycling processes, and the sustainability of the materials after multiple recycling cycles?

10.   The economic feasibility of utilizing these polymeric biocomposites is briefly discussed. It would be valuable to provide a more detailed analysis of the cost-effectiveness, considering factors such as raw material availability, production scalability, and potential market applications.

11.   The presented research focuses on utilizing waste from the footwear industry. Are there any limitations or challenges associated with obtaining a sufficient and consistent supply of the required waste materials? How do these limitations impact the feasibility of large-scale production?

12.   The biocomposites are evaluated for their potential use in the automotive industry. discuss the specific automotive applications where these materials could be beneficial and the potential advantages they offer over traditional materials?

13.   The authors mention the use of the biocomposites as a sustainable alternative for traditional materials. Have there been any preliminary studies or comparisons made regarding the environmental impact and sustainability aspects of these biocomposites in comparison to conventional materials used in similar applications?

14.   The conclusions highlight the potential of the polymeric biocomposites for various applications. Are there any ongoing or future research directions that authors are pursuing to further enhance the properties or expand the range of applications for these materials?

Minor editing of English language required.

Author Response

Thank you for your letter and comments on our manuscript titled "Polymeric Biocomosites Based on Thermoplastic Polyurethane (TPU) and Protein and Elastomeric Waste Mixture, ID MANUSCRIPT - 2488139.

The main comments and our specific response are detailed below. Please see the attachment.

Best regards!

Prof. Phd. eng. Oprea Ovidiu

Phd Eng Nituica Mihaela

Reviewer 2 Report

The paper addresses a topic of interest and focuses on a very current theme, namely circular economy; It is well-structured and balanced in its various parts, making it understandable to the reader.
The experiments are described satisfactorily both in terms of sample preparation and analytical techniques. The conclusions support the results.
The only aspect, perhaps overlooked in a context where analytical techniques are widely used, which would have complemented the work, is a compositional analysis of the elemental samples obtained. The reason may be to ensure the absence of metals or other substances that could be potentially harmful to health.
However, this is merely a consideration to ensure that the materials that may be used in footwear production are not risky for health.
There are no corrections to report, except in Table 3, where it is necessary to review the significant figures of the T20 value.

Author Response

Thank you for your letter and comments on our manuscript, ID manuscript -materials 2488139, 

The main comments and our specific response are detailed below. Please see the attachment.

Best regards!

Prof. PhD eng Ovidiu Oprea

PhD eng Nituica Mihaela

Round 2

Reviewer 1 Report

the revised manuscript can be accepted.